# Manifestation of Triploid Heterosis in the Root System after Crossing Diploid and Autotetraploid Energy Willow Plants

**DOI:** 10.3390/genes14101929

**Published:** 2023-10-12

**Authors:** Dénes Dudits, András Cseri, Katalin Török, Radomira Vankova, Petre I. Dobrev, László Sass, Gábor Steinbach, Ildikó Kelemen-Valkony, Zoltán Zombori, Györgyi Ferenc, Ferhan Ayaydin

**Affiliations:** 1Institute of Plant Biology, HUN-REN Biological Research Centre, 6726 Szeged, Hungary; dudits.denes@brc.hu (D.D.); torok.karolyne@brc.hu (K.T.); sass.laszlo@brc.hu (L.S.); zombori.zoltan@brc.hu (Z.Z.); 2Institute of Experimental Botany, Czech Academy of Sciences, 165 02 Prague, Czech Republic; vankova@ueb.cas.cz (R.V.); dobrev@ueb.cas.cz (P.I.D.); 3Laboratory of Cellular Imaging, HUN-REN Biological Research Centre, 6726 Szeged, Hungary; steinbach.gabor@brc.hu (G.S.); kelemen.ildiko@brc.hu (I.K.-V.); ayaydin.ferhan@brc.hu (F.A.); 4Hungarian Centre of Excellence for Molecular Medicine (HCEMM) Nonprofit Ltd., 6728 Szeged, Hungary

**Keywords:** *Salix*, hybrid vigor, mid-parent heterosis, root development, auxin–cytokinin ratio, cell cycle

## Abstract

Successful use of woody species in reducing climatic and environmental risks of energy shortage and spreading pollution requires deeper understanding of the physiological functions controlling biomass productivity and phytoremediation efficiency. Targets in the breeding of energy willow include the size and the functionality of the root system. For the combination of polyploidy and heterosis, we have generated triploid hybrids (THs) of energy willow by crossing autotetraploid willow plants with leading cultivars (Tordis and Inger). These novel *Salix* genotypes (TH3/12, TH17/17, TH21/2) have provided a unique experimental material for characterization of Mid-Parent Heterosis (MPH) in various root traits. Using a root phenotyping platform, we detected heterosis (TH3/12: MPH 43.99%; TH21/2: MPH 26.93%) in the size of the root system in soil. Triploid heterosis was also recorded in the fresh root weights, but it was less pronounced (MPH%: 9.63–19.31). In agreement with root growth characteristics in soil, the TH3/12 hybrids showed considerable heterosis (MPH: 70.08%) under in vitro conditions. Confocal microscopy-based imaging and quantitative analysis of root parenchyma cells at the division–elongation transition zone showed increased average cell diameter as a sign of cellular heterosis in plants from TH17/17 and TH21/2 triploid lines. Analysis of the hormonal background revealed that the auxin level was seven times higher than the total cytokinin contents in root tips of parental Tordis plants. In triploid hybrids, the auxin–cytokinin ratios were considerably reduced in TH3/12 and TH17/17 roots. In particular, the contents of cytokinin precursor, such as isopentenyl adenosine monophosphate, were elevated in all three triploid hybrids. Heterosis was also recorded in the amounts of active gibberellin precursor, GA_19_, in roots of TH3/12 plants. The presented experimental findings highlight the physiological basics of triploid heterosis in energy willow roots.

## 1. Introduction

Short rotation plantations of the fast-growing trees, such as energy willow (*Salix* spp.), can play a key role in combating climate change and improving energy security. Maximizing the biomass yield requires integrated contributions from willow breeding and agrotechnology, which are both supported by plant research. Shrub willows grown as a woody crop have outstanding potential to serve as an optimal feedstock to produce bioenergy, biofuels, and bioproducts with environmental and rural development benefits [1]. As wood pellets, it can be used in gasifier to produce syngas [2,3]; alternatively, it can serve as raw material for cellulosic ethanol production [4,5]. In addition, the energy-willow-based pyrolysis system for biomethane production led to increased energy performance and negative global warming potential [6]. Recent studies by Kakuk et al. [7] demonstrated that green willow biomass harvested as less than one-year-old shrubs could be efficient biogas substrate compared to woody tissues.

Out of serious global ecological problems, the pollution of the soil with toxic heavy metals or with organic pollutants urges application of phytoremediation as a cheap and environmentally sustainable method for land decontamination, preferentially based on woody species [8,9,10]. Use of willow genotypes capable of heavy metal and organic phytoremediation associated with biomass production gains an increasing interest world-wide [11,12]. Willow cultivation on saline-alkali soil lands is dependent on the salt tolerance of Salix species [13].

The environmental and economical sustainability of the above-listed multipurpose applications exploits some key properties of energy willow genotypes that can be generated by improved breeding technologies (see the review by Hanley and Karp, [14]). The increase in the biomass yield and energy production is a central goal [15]. Willow breeding can rely on the extended biodiversity of the genus *Salix*, which includes 330–500 species and more than 200 hybrids [16]. In the genus *Salix*, the formation of new species involved hybridization and subsequent polyploidization [17]. Crossing between *Salix miyabeana* as a tetraploid species native to Japan, Korea, and China and diploid *S. purpurea* or its hybrids resulted in triploid genotypes with higher green biomass yield than their diploid or tetraploid parental species [18]. In these studies, the highest-yielding genotype was a triploid hybrid (*S. koriyanagi* × *S. purpurea*) × *S. miyabeana*. These interspecific triploid hybrids displayed significant heterosis for the harvestable biomass and biomass-related growth traits in the greenhouse and in the field [19]. Cheng et al. [20] reported a large number of differentially expressed genes by comparing synthetic *Populus* allotriploid (2n = 3x = 57) and their diploid parents (2n = 2x = 38). These allotriploid plants exhibited higher heights.

Considering the above outlined extensive applications of energy willow plants grown as short rotation coppice for biomass production or phytoremediation, it is evident that the root system parameters can play a key role in the efficiency of these activities. Improvement of the root characteristics is a special breeding target that can be supported by phenotyping technologies (reviewed by Tracy et al. [21]). In only a few cases, heterosis was recorded in root growth characteristics. Hund et al. [22] reported the expression of marginal overdominance with respect to different traits of root system growth in maize. The relative Mid-Parent Heterosis for maize root traits was higher under stress generated by low phosphorus availability [23]. Natural or artificial multiplication of the plant genome size (that is, polyploidization) can also cause essential alterations in different traits of the plant anatomy and functions or even in the stress responses (see review by Ruiz et al. [24]). The interplay between ploidy and heterosis was concluded from the studies on triploid maize hybrids. Yao et al. [25] showed that the heterosis is subject to dosage effects. In contrast, the paternal genome dosage in F1 triploid hybrids of sugar beet does not enhance heterosis effects beyond what can be achieved in F1 diploid hybrids [26]. The progenies from the crossings between the diploid *Camellia sinensis* and tetraploid variants showed heterosis for caffeine and epigallocatechin-3-gallate contents [27].

Previously, we reported the production and the detailed characterization of a set of autotetraploid *S. viminalis* var. Energo genotypes (polyploid Energo [PP-E]; (2n = 4x = 76, [28]). Comparison of diploid and tetraploid variants revealed complex, multiple changes at the anatomical and morphological levels and in growth parameters of energy willow organs. Duplication of the genome size resulted in significant stimulation of root development. In parallel, we showed an improved salt tolerance of tetraploid energy willow genotypes [29]. As an extension of our energy willow breeding program, we crossed leading Swedish diploid cultivars (Tordis and Inger) with our autotetraploid genotypes. The detailed characterization of the aboveground traits clearly indicated the expression of Mid-Parent Heterosis (MPH%) in the shoot growth rate or in the CO_2_ assimilation rate [30]. The present paper is focused on the characterization of triploid heterosis controlling the biomass, growth rate, cellular parameters, and hormonal status of roots. The presented data provide deeper insight into how root traits can be improved through the combination of hybrid vigor with polyploidization.

## 2. Materials and Methods

### 2.1. Crossing Diploid Tordis or Inger Female Plants with Autotetraploid Male Plants and Propagation of Triploid Energy Willow Genotypes

Previously, we produced autotetraploid (2n = 4x = 76) energy willow plants and described in detail the main characteristics generated by genome duplication [28]. Parental stem cuttings were collected in wintertime and dormancy was released in water under greenhouse conditions. Male catkins from the tetraploid PP-E7 and PP-E15 plants were removed at anthesis for the collection of pollen to accomplish artificial pollination; pollen was applied on the receptive stigma of Tordis and Inger cultivars. After the isolation of 10- and 14-day-old hybrid embryos, they were cultured on the half MS hormone-free medium. The in vitro grown plants were rooted and transferred to soil and propagated by stem cuttings to be used in phenotyping studies.

### 2.2. Flow Cytometry

Determination of ploidy levels of obtained hybrids was conducted by flow cytometry. Root tips (approximately 5–7 mm) were excised from the plants grown in Knop’s solution: (Ca(NO_3_)_2_: 0.55 mM; KNO_3_: 1.90 mM; NaCl: 0.10 mM; KH_2_PO_4_: 0.40 mM; MgSO_4_: 0.50 mM; and FeCl_3_: 3.8 µM, pH of 5.5). Determination of ploidy levels was conducted by flow cytometry (BD FACSCalibur) equipped with a 532 nm green solid-state laser operating at 30 mW, as we published before [28]. The suspension of released nuclei was stained with 1 μg mL^−1^ propidium iodide (Sigma, Livonia, MI, USA) for 10 min. The cell cycle phase distribution was analyzed using ModFit LT™ version 3.0 (Verity Software House, Toshan, ME, USA).

### 2.3. Phenotyping of Triploid Heterosis in Root Growth in Soil

Single dormant stem cuttings were planted into radio-tagged Plexiglas columns (100 mm in diameter and 400 mm deep) with a mixture of 80% Terra peat soil and 20% sandy soil. Each genotype was represented by 10 replications. The Plexiglas columns were photographed from four different side positions and from the bottom [28]. The root-related white pixels were identified by subtracting the black soil background from the images. To characterize the root surface area (cm^2^) appearing at the surface of the chamber, the metric values of the areas of the four side view projections (90° rotation) were summarized, and the metric value of the area of the bottom view was added.

After completion of the 8-week phenotyping experiment, the rooted stems were removed from the soil, and the roots were separated from the soil. The root weights were measured immediately after removal from the soil (fresh weight).

Mid-Parent Heterosis (MPH) values were calculated as heterosis over mid-parent (MPH%) = [(F1 − MP)/MP × 100], where F1 is the numerical value trait measurement in the hybrid and MP values are the mean values of the parents (P1 + P2)/2.

### 2.4. Quantification of Triploid Heterosis by Analyses of Root Growth Intensity Parameters in Knop’s Solution

Stem cuttings (10 cm) of triploid hybrid and parental genotypes were placed into holes of plastic foam floating in a pool filled with Knop’s solution. The length of the main roots was measured in the most developed four root systems from each genotype represented by three cuttings. For determination of root growth intensity, we measured the length of the same main roots twice in 48 h intervals. We also estimated heterosis over the cultivar parent (CPH%) = [(F1 − CP)/BP × 100].

### 2.5. Root Sections, Microscopy, and Measurement of Cell Size

Stem cuttings were rooted in Knop’s-solution-saturated zeolite-containing jars. The roots were harvested when they reached more than 5 cm in length. Slices of disks were cut manually under a stereozoom microscope at the beginning of vasculature at the root tip (between the end of the division zone and the beginning of the elongation zone). The disks were mounted with 10 × 10 mm square coverslips using phosphate-buffered saline (PBS) onto standard glass microscope slides. Cell wall autofluorescence was captured using 405 nm laser excitation and a wide spectral emission detection between 415 nm and 602 nm detection of Leica SP5 AOBS confocal laser scanning microscope (Leica Microsystems GmbH, Wetzlar, Germany). A 40× dry objective (HCX PL Fluotar, N.A 0.75, Leica Microsystems CMS GmbH, Wetzlar, Germany) was used with pinhole (set to 1 Airy unit (113 µm). Bright field images were captured with a transmitted detector using the same laser line. The diameters of the root parenchyma cells were measured using FIJI software (v1.53c). Two concentric circles (0.33r and 0.83r, r = radius of the disc section) were overlaid onto the root disk image to exclude heterogeneous population of cells in the central vascular area and epidermal region. Diameters of ~50 parenchyma cells (per disk) were recorded within the area delineated by the two overlaid circles using a macro in FIJI [31]. The macro written for the FIJI program (v1.53c) was used to establish the initial settings of the image imported in Leica LIF file format. The macro commands are the following. 1. Select fluorescence channel, 2. Revert grayscale image to cyan LUT, and 3. Set the line tool for the measurement. The physical lengths of the selected lines were determined based on the scale imported from the microscope from the LIF file. The macro allowed streamlining of the measurement protocol. We measured the diameters of a total of 1649 root cells from 6 different lines. Charts were prepared after exporting of the measurements to Microsoft Excel 2010 (Microsoft, Redmond, WA, USA). Composite images were prepared using CorelDraw Graphics Suite X7 (Corel Corporation, Ottawa, ON, Canada)

### 2.6. Analysis of Hormone Contents in Main Root Tips

Root tissue samples (30–80 mg fresh weight) were purified and analyzed according to Dobrev and Kamínek [32] and Dobrev and Vankova [33], as described in our previous publication [28]. The root tip tissues were collected in two replications, and the mean values are provided. Briefly, samples were, after homogenization, extracted with cold (−20 °C) methanol:water:formic acid (15:4:1, *v*/*v*/*v*). The labelled internal standards (10 pmol per sample) were added. The phytohormones were separated with a mixed-mode reverse phase cation-exchange SPE column (Oasis-MCX; Waters, Milford, MA, USA) into the acid fraction by elution with methanol (auxins, ABA, SA, JA, GA) and into the basic fraction by elution with 0.35 M NH_4_OH in 60% methanol (CKs). Hormone metabolites were analyzed using HPLC (Ultimate 3000; Dionex, Sunnyvale, CA, USA) coupled to a hybrid triple quadrupole/linear ion-trap mass spectrometer (3200 Q TRAP; Applied Biosystems, Foster City, CA, USA).

### 2.7. Statistical Analyses

In the statistical analyses, the significance of differences among genotypes was evaluated using Student’s *t*-test for comparisons between the traits of diploid parents and triploid hybrids. Statistically significant events in the traits compared with the triploid hybrids’ mean values are indicated with mean labels as *p* ≤ 0.01 ***, *p* ≤ 0.05 **, and *p* ≤ 0.1 *.

## 3. Results

### 3.1. Generation of Triploid Hybrids by Crossing Autotetraploid and Diploid Energy Willow Plants

Cuttings of woody stems of willow genotypes were rooted in water and used for nuclei isolation from root tips. DNA ploidy levels were screened by flow cytometry for identification of the triploid lines. Typical output histograms of the flow cytometry for selected hybrids and their parents studied in the present work are shown in Figure 1. Based on these DNA content indicators, we have established a collection of triploid hybrid (TH) lines for studies on the expression of hybrid vigor in root traits.

### 3.2. Expression of Triploid Heterosis in Root Biomass as Shown by Root Phenotyping of Willow Plants Grown in Soil

Previously, we reported the use of a root phenotyping platform for characterization of the willow root system [28]. As shown in Figure 2, significant differences in root formation can be recorded between the parental plants and their triploid hybrids. The two triploid hybrids (TH3/12 and TH21/2) developed an enlarged root system in comparison to their parental plants. Differences in root formation are more recognizable on pictures from the bottom side.

To quantify genotype-dependent root biomass production, Table 1 presents the total surface area values (in cm^2^) occupied by white pixels and root wet weights. In each crossing combination, we have calculated Mid-Parent Heterosis (MPH) % values for comparison of heterosis levels.

In the present experimental design, we could compare heterosis effects in hybrids with different parental combinations. In the root surface area parameter, we recognized significantly different responses after crossing the Swedish Tordis male cultivar with the PP-E7 or the PP-E15 tetraploid plants. High triploid heterosis was expressed in the TH3/12 plants (female parent PP-E7), while the TH17/17 plants (female parent PP-E15) failed to exhibit a positive heterosis response. The PP-E7 plants as crossing partners could also ensure heterosis in other crossing combinations, as with the Inger plants. In the case of TH21/2 plants, the MPH% was lower, which is possibly due to the larger surface area of the root system of Inger parental plants compared to Tordis plants. After eight weeks of the growing period, the roots were separated from the soil, and the root biomasses were measured as fresh weights (Table 1). This parameter has indicated MPH heterosis in all three hybrid combinations, but with lower values as detected by the root surface area data shown by phenotyping, even in the case of the TH17/17 roots. This difference in recording by phenotyping and root weight measurement of the root biomass may originate from the limitation of digital imaging of roots grown in Plexiglas columns. Fresh weight values for the TH plants were close to tetraploid parental roots and exceeded the root formation of diploid parents. Subsequently, the MPH% values were found to be low in this trait. However, the fresh root weight of TH21/2 hybrid plants exceeded the root weights of both parental genotypes. The average increase reached 10% even when compared to the PP-E7 tetraploid plants. This difference was found to be significant.

To gain more insight into the dependence of root development on the genotype of willow plants, we have recorded the root surface area values during eight weeks of the growing period (Figure 3). The sum of the side and bottom images of the Plexiglas columns indicated root biomass differences after four weeks.

The tetraploid PP-E7 plants showed lower root surface area during almost the whole growing period. After four weeks, the diploid plants of the Tordis cultivar also formed less roots in comparison to the triploid hybrids. The TH21/2 triploids produced the highest root surface area values, especially during the last two weeks of culture. Out of parental lines, Inger plants have good root formation potential.

### 3.3. Deeper Analysis of the Triploid Hybrid Vigor in Root Growth Rate In Vitro

In attempt to gain more information about the expression of triploid heterosis in root characteristics, willow stem cuttings were rooted in Knop’s solution in a thermostat room. As shown in Table 2, the rooting parameters varied significantly during the early phase of root formation. After 16 days of the growing period, the root length of parental diploid Tordis plants was longer than that of Inger plants.

Under these experimental conditions, the tetraploid stem cuttings produced the longest primary roots, especially in the case of the PP-E7 plants. Characterization of this trait indicated an advantage of hybridity expressed in root growth of the TH21/2 plants. The Mid-Parent Heterosis (30.67%) was detectable in this triploid hybrid combination. As a common trend, we recorded the fastest growth rate in the roots of tetraploid parental plants. This feature could contribute to the low or the zero MPH% values recorded in these crossing combinations. Because in the use of the triploid hybrids for bioenergy production or phytoremediation any hybrid will be compared to commercial cultivars, we present the heterosis values relative to the parent cultivars (CPH%). Taking this into consideration, we can postulate that all of the triploid hybrids, especially the TH3/12 plants, showed significant hybrid vigor.

To better understand the physiological backgrounds of triploid heterosis, we have determined frequencies of root cells in various cell cycle phases. The data generated by flow cytometry failed to show a general tendency, except in the ratio of the G2/M phase cells (Table 2). These cell cycle phases were represented with the lowest frequency in the tetraploid root cells. In the triploid hybrid roots, the occurrence of G2/M cells showed intermediate frequency values between the parental genotypes with zero MPH%.

### 3.4. Comparison of Cellular Structures in Roots of Triploid Hybrids and Their Parents

To assess the influence of triploid hybrid vigor at the cellular and tissue levels, cell wall autofluorescence was imaged by laser scanning confocal microscopy on hand-sectioned, unfixed root tips (Figure 4).

In general, the root tissue architecture of the analyzed triploid (TH17/17, TH21/2) and tetraploid (PP7, PP15) plants was comparable to that of diploid Inger and Tordis lines, with a well-defined stele, cortex (parenchyma), and epidermis without notable abnormalities (Figure 4b–g). On the other hand, a marked increase in cell size was discernable in the root cortex of triploid lines during microscopy observations. The root cortex contains multiple files of parenchyma cells, which play an important role in determining the root thickness. Cell size measurements showed that roots of the TH17 and the TH21 triploid plants contain larger average parenchyma cell diameter compared to their diploid and tetraploid parents, indicating a clear heterosis effect at the cellular level (Figure 4).

### 3.5. Hormonal Background of Growth Responses in Roots of Triploid Hybrids and Their Parents in Knop’s Solution

It is well established that plant hormones are involved in the regulation of root development. Therefore, determination of the actual concentration of these compounds may shed light on the molecular factors controlling triploid heterosis in energy willow. In the present study, the rooting of stem cuttings from the hybrids and their parents offered a unique experimental system to screen concentration differences in key growth regulators as factors with potential involvement in the generation of hybrid vigor. Quantification of individual plant hormones resulted in a complex picture with differences between various hybrid combinations (Table 3).

The interpretation of these data should be linked to growth intensity data presented in Table 2 showing a fast-growing rate for the tetraploid parents and the triploid plants with Mid-Parent Heterosis (TH3/12:70.08%; TH21/2 10.17%). The TH3/12 plants presented significant MPH% in levels of CK precursors, especially isopentenyl adenosine monophosphate and gibberellin precursor GA19. Furthermore, accumulation of auxin and ABA deactivation metabolites, such as IAA-glutamate, or phaseic acid were characteristic of roots of the TH3/12 plants with only moderate MPH% in amounts of the *trans*-zeatin riboside monophosphate and of ABA catabolite dihydrophaseic acid. Reduced concentration of indole-3-acetic acid, salicylic acid, as well as CK deactivation metabolites CK *O*-glucosides was recorded in TH3/12 hybrids relative to their parents. We could identify TH3/12 plants with intermediate values in contents of phenylacetic acid, benzoic acid, and jasmonic acid.

The TH21/2 triploid hybrid roots with low MPH% value in growth rate showed high MPH% in the concentration of IAA-glutamate and in total cytokinin content. In the case of dihydrophaseic acid and phaseic acid, the MPH% values were low. In these roots, reduced concentrations of salicylic acid, jasmonic acid, abscisic acid, and CK *O*-glucosides were detected in comparison to the parental roots. A considerable fraction of hormones represented intermediate amounts relative to the parental roots. The TH17/17 triploid hybrids from crossing between Tordis and PP-E15 plants failed to express MPH% in root growth intensity. Similarly, several hormones exhibited low or intermediate amounts, except CK ribosides and CK phosphates (i.e., *trans*-zeatin riboside monophosphate, isopentenyl adenosine monophosphate, and *cis*-zeatin riboside monophosphate) with MPH varied between 2.26% and 21.33% (see Table 3).

## 4. Discussion

It is evident that plant root systems exhibit a multitude of important roles both in the control of aboveground biomass production and in the efficiency of phytoremediation. Therefore, enhancing the capacity of root functions determined by several physiological parameters is also a key breeding target. The utilization of natural and artificial genetic variability for root trait improvement is more efficient when performed through the application of root trait phenotyping (see review by Tracy et al. [21]). Out of several breeding techniques, we previously reported that the autotetraploidization of energy willow plants from the Hungarian cultivar, Energo, significantly stimulated root development, with a simultaneous increase in the amounts of indole acetic acid, cytokinins, gibberellin, and salicylic acid in the root tips of these plants [28]. This artificially generated genetic stock opened an avenue for further breeding of triploid genotypes by crossing between these tetraploid willow plants with leading diploid Swedish cultivars [30]. Previously, Serapiglia et al. [18] published that triploid species hybrids (*S. koriyanagi* × *S. purpurea*) × *S. miyabeana* produced the highest biomass yield under field conditions. Later, these high-yielding genotypes were selected for stable performance [19]. An artificial triploid Populus family was published after hybridization between *Populus simonii* × (*P. nigra var. Italica*) × (*P. popularis*) species. Zhang et al. [34] reported a “Gigas” effect on leaf size. Populus allotriploid plants (2n = 3n = 57) were generated from the hybridization of induced 2n female gametes of *Populus pseudosimonii* × *P. nigra* and *P.* × *beijingensis* [35]. These poplar plants showed more vigorous growth and more efficient photosynthesis, carbon fixation, sucrose, and starch synthesis. None of these cited research reports provided information about the expression of heterosis in root traits, particularly in woody plants. In the present program, the triploid heterosis was analyzed in hybrid plants from commercial cultivars with a focus on the root system.

### 4.1. Control of the Size of the Root System

Cultivation of energy willow plants in the Plexiglas columns allowed for the production of two types of parameters for the quantification of root biomass. The root surface area data were generated by a phenotyping platform determining root-related pixels. Table 1 presents these values collected at the end of a growing period of eight weeks in comparison to root fresh weight values measured after separation of root tissues from the soil. Both approaches have a margin of error because digital images did not record roots deep inside the columns; therefore, we also determined the root surface areas at the bottom surfaces. We also have to consider that the separation of roots from the soil cannot be complete. Despite these limitations, the comparison of genotypes clearly indicated Mid-Parent Heterosis to a varying degree. According to the root surface area values, hybrids (TH3/12 and TH21/2) from crossings between PP-E7 tetraploid plants and Tordis and Inger showed clear hybrid vigor (MPH 43.99% and MPH 26.93%) compared to the average of both parents. In contrast, progeny plants of PP-E15 tetraploid parent (TH17/17) failed to produce larger root systems than their parental genotypes. The ranking of genotypes according to the root surface area or the fresh weight values showed similar trends (Table 1). The fresh weight data also showed heterosis, but with lower MPH% values. According to this parameter, the TH17/17 hybrid plants also express low hybrid vigor. Differences in the level of combining ability between crossing partners can have a direct effect on heterosis [36]. In this respect, we may suggest that the tetraploid PP-E7 plants are more optimal partners than plants of PP-E15 tetraploid genotypes in the generation of triploid heterosis.

The root initiation of stem cuttings and the early root development showed significant differences between various willow genotypes, as indicated by the root surface area data monitored by phenotyping. Figure 3 presents root growth curves for two hybrids expressing heterosis (TH3/12 and TH21/2) in addition to their parental plants. Differences between genotypes are increased after six weeks, except for the tetraploid PP-E7 parental plants, which showed delayed rooting. However, these plants present larger root surface area values than plants of the Tordis cultivar at the termination of the experiments. The rooting vigor is significantly higher using Inger plants as crossing partners than the hybridization carried out with plants of the Tordis cultivar. This feature is expected to contribute to the hybrid vigor of TH21/2 hybrids showing the highest root biomass (Table 1). In contrast, the MPH% value indicates enhanced triploid hybrid heterosis for the TH3/12 plants, which are the progenies of the Tordis cultivar with a moderate root system. Because both hybrids are progenies of the tetraploid PP-E7 plants, we can propose that the differences might be due to the combination of these two diploid cultivars.

### 4.2. Primary Factor in Generation of Triploid Heterosis in Root Systems

To obtain deeper insight into the characteristics of root system, the use of in vitro cultures can be a prerequisite [22,37]. In the present study, willow cuttings were rooted in Knop’s solution for quantification of early growth intensity for the triploid hybrids and their parents. Table 2 provides root length data as a reference for the developmental stage during the last two days of measurement. In all three crossing combinations, the tetraploid parents developed the longest roots. Under the culture conditions used, the stem cuttings of TH3/12 showed delayed rooting and reduced early growth. The Mid-Parent Heterosis of 28.46% could be recorded in the root system of TH21/2 hybrid plants. Root growth rates showed opposite trends when we compared plants from the commercial cultivars and the tetraploid parents. For two days, roots of the PP-E7and the PP-E15 genotypes produced 35–38 mm growth, while roots of Inger and Tordis plants grew only 13–18 mm. The MPH value (70.08%) reflected hybrid vigor only in the TH3/12 plants. This response is in good agreement with the highest heterosis in the root surface area value of the TH3/12 plants in soil (Table 1). The interpretation of the data of Mid-Parent Heterosis in the triploid willow hybrids needs to consider the high growth intensity of the tetraploid parents. Even though the Mid-Parent Heterosis % is the most used parameter for the presentation of hybrid vigor, we also used growth intensity data of commercial cultivars as a reference. As shown in Table 2, significant hybrid vigor can be seen if the Cultivar-Parent Heterosis (CPH%) is analyzed. In this regard, all triploid willow hybrids showed heterosis, with the highest root growth rate in the case of TH3/12 plants.

Increase in the root length during growth is dependent on the cell division rate in the meristematic region and elongation in the differentiation zone (reviewed by Perilli et al. [38]). In attempt to discover the cellular basis of different root growth rates, we also used flow cytometry to identify frequencies of root tip cells at various cell cycle phases. Table 2 presents only the ratio of G2/M cells, because we could detect a general trend only in this parameter during comparison between triploid hybrids and their parents. A high number of G2/M cells was characteristic for roots of the diploid parental cultivars with low root growth intensity, while the tetraploid roots contained a low number of G2/M cells during the fast growth. In maize roots, the frequency of cells in the G2 phase significantly increased in the elongation zone [37]. It is possible that the high growth rate in tetraploids may originate from the more extended elongation zones. In the hybrid roots, the numbers of G2/M cells are intermediate between the parental values, with an intermediate growth intensity rate, except for TH3/12 roots. The triploid hybrid vigor could be seen in the size of the parenchyma cells in the root cortex. This result is in accordance with earlier observations that in maize hybrid leaves, overdominance increased the cell length and width [39].

### 4.3. Dependence of Triploid Heterosis on a Shift in the Auxin–Cytokinin Ratio in Willow Roots

Assessing a possible link between root growth and plant hormones can highlight molecular regulatory elements in triploid heterosis observed in the energy willow root system. As reviewed by Saini et al. [40], the crosstalk between auxin and other plant hormones is a central mechanism in the regulation of root behavior. Considering the level of triploid heterosis in the root surface area (cm^2^), the TH3/12 and TH21/2 plants showed increases relative to both parental genotypes. In the growth intensity, heterosis was detectable compared to the diploid parental cultivars. In the present study, 5–7 mm sections of root tips representing meristematic and short elongation zones were collected for hormone analyses. The strongest heterosis and most significant increase in hybrid vigor were found in the roots of TH3/12 plants. The contents of the indole-3-acetic acid and the phenylacetic acid were lower in these F1 primary roots than in the roots of parents (Table 3). In addition, significant heterosis was realized in the elevation of the IAA-glutamate contents, which may indicate enhanced IAA turn-over. Auxin conjugation to glutamate produces indole-3-acetyl glutamic acid (IAAGlu) as an inactive product (see review by Korasick et al. [41]). In 10-day-old seedlings of *Theobroma cacao* (cacao), a sharp reduction in IAA concentration and an increase in IAA-Glu content was reported, especially in meristematic and elongation zones [42].

The low auxin levels in the primary roots of triploid hybrids TH3/12 and TH17/17 can alter the auxin–cytokinin ratio, given that high Mid-Parent Heterosis values were recorded for cytokinin precursors (namely, cytokinin phosphates and their ribosides) (see Table 3). In root tips from Tordis plants, the auxin content (IAA + phenylacetic acid) was seven times higher than the sum of the cytokinin metabolites. This ratio was essentially lowered in the PP-E7 (1.90) and the PP-E15 (4.84) tetraploids roots. In triploid hybrid progenies, even more cytokinins were detected relative to the auxin contents. The auxin–cytokinin ratios were 1.68 for the TH3/12 root and 3.31 for the TH17/17 root. In root tips from diploid plants of the Inger cultivar and its TH21/2 triploid hybrids, the auxin–cytokinin ratios were 3.21 and 2.89, which also indicate the dominant role of cytokinins. Antagonistic auxin and cytokinin interaction plays a critical role in the regulation of root meristem development (see reviews by Su et al. [43]). Auxin is required for meristem cell division, while cytokinin antagonizing auxin signaling in the transition zone affects cell differentiation and elongation [44]. Stimulation of cytokinin signaling via receptor AHK3 and type-B response regulator represses auxin signaling. Positive cytokinin-response regulators ARR1 and ARR12 promote the expression of the *SHORT HYPOCOTYL 2* (*SHY2/IAA3*) gene in the vascular tissues of roots in the transient zone. The SHY2 protein acts as a repressor of auxin signaling through negative regulation of the PINs as the auxin transporters. As such, cytokinins cause auxin redistribution, prompting cell differentiation [45]. In the present study, the lower auxin–cytokinin ratios could be responsible for the larger cell size in roots of triploid hybrids with heterotic responses, which may explain the observation of considerably thicker roots in triploid hybrids (Figure 2 and Figure 4c–f).

Elevated contents of the isopentenyl adenosine monophosphate (iPMP) indicated heterosis in all three triploid hybrids, with the highest MPH % in roots of TH3/12 hybrids (Table 3). The iPMP is a cytokinin precursor, which can be hydroxylated to *trans*-zeatin riboside-5-monophosphate. Both precursors (isopentenyl adenosine monophosphate and t*rans*-zeatin riboside monophosphate) provide active cytokinin bases either directly through LOG enzyme catalysis or via riboside formation [46]. Accumulation of the *cis*-zeatin riboside monophosphate in the roots of triploid hybrids also indicates heterosis (Table 3). In the roots of TH3/12, considerable heterosis was recorded in the amount of active gibberellin precursor GA19 (Table 3), which can further support the central role of increased cell size in root heterosis. In Arabidopsis roots, the gibberellin-deficient mutants showed shorter root length, reduced meristem size, and smaller cell length in comparison to the wild-type roots [47]. In root tips of the two triploid hybrids (TH3/12 and TH21/2), contents of abscisic acid (ABA) and its metabolites (dihydrophaseic acid, phaseic acid) reflect the variable degree of heterosis (Table 3). Especially high MPH % was correlated with the amount of phaseic acid, the primary ABA catabolite. In Arabidopsis, the biphasic effect of ABA included the stimulatory impact. Exogenously applied 0.1 μM ABA caused a 30% faster root growth rate [48].

The triploid heterosis detected in root growth parameters can be a consequence of a reduction in the amount of growth-inhibitory hormones salicylic acid and jasmonic acid in comparison to the parental roots. The benzoic acid levels showed intermediates in the triploid hybrid roots (Table 3). The salicylic acid can influence root meristem patterning in a concentration-dependent manner [49]. Inhibition of root elongation by jasmonic acid via a reduction in the cell number and size was reported (see review by Xu et al. [50]).

## 5. Conclusions

In the present study, we integrated polyploidy and heterosis effects through the production of triploid hybrids for detailed analysis of consequences in the root development of energy willow plants. Because root functions play a pivotal role in biomass productivity and in phytoremediation, the presented data can support an extended use of this energy plant in a short rotation cultivation system. The use of a phenotyping platform allowed us to characterize the root parameters of the triploid hybrids and their parents, while the in vitro culture in Knop’s solution provided root samples for monitoring the growth rate, cell cycle events, and differences in hormonal status of roots. The Mid-Parent Heterosis (MPH%) in the root biomass yield showed variation between crossing combinations, with an essential contribution from the tetraploid parents having enlarged root system. In particular, differences in the growth rate were significant between roots from the diploid cultivars and the tetraploid genotypes. Therefore, we provide Cultivar-Parent Heterosis (CPH%) values in all three crossing combinations. The search for the hormonal background of triploid heterosis highlighted the role of several plant hormones and their ratios.

The reduction of the auxin–cytokinin ratios in the roots of triploid hybrids indicated a dominant role for cytokinins. The hormone analysis revealed heterosis in additional hormones. These findings emphasize a complex regulation in the expression of the triploid heterosis during the formation of the willow root system. The hormonal influences in MPH% can be responsible for alterations at the cellular level. The enlarged root system in defined crossing combinations (TH3/12 and TH21/2 triploid hybrids) can have a significant role in biomass yield, drought tolerance, or the efficiency of phytoremediation. Therefore, these genotypes can be integrated in energy willow breeding programs.

## Figures and Tables

**Figure 1 genes-14-01929-f001:**
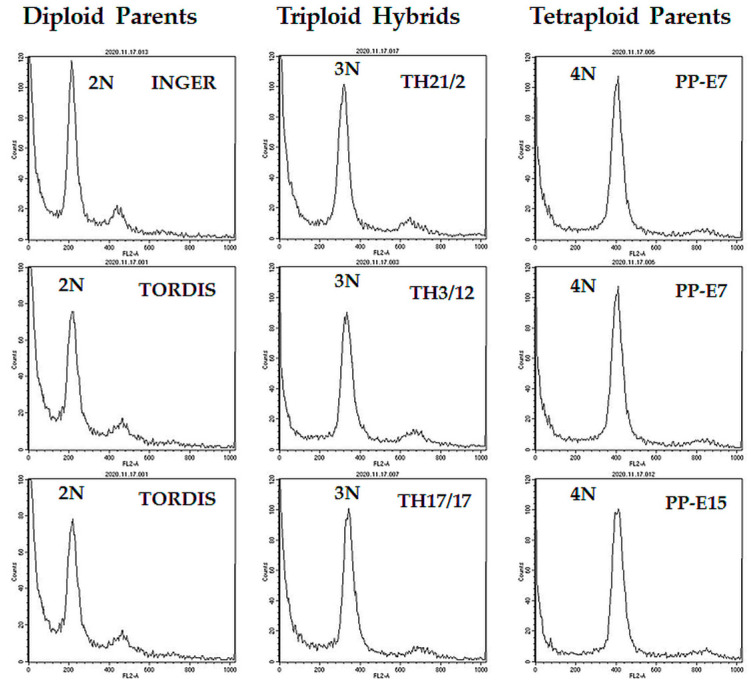
Identification of triploid willow hybrids by flow cytometry analysis of relative DNA content in root tip cells (using propidium iodide staining).

**Figure 2 genes-14-01929-f002:**
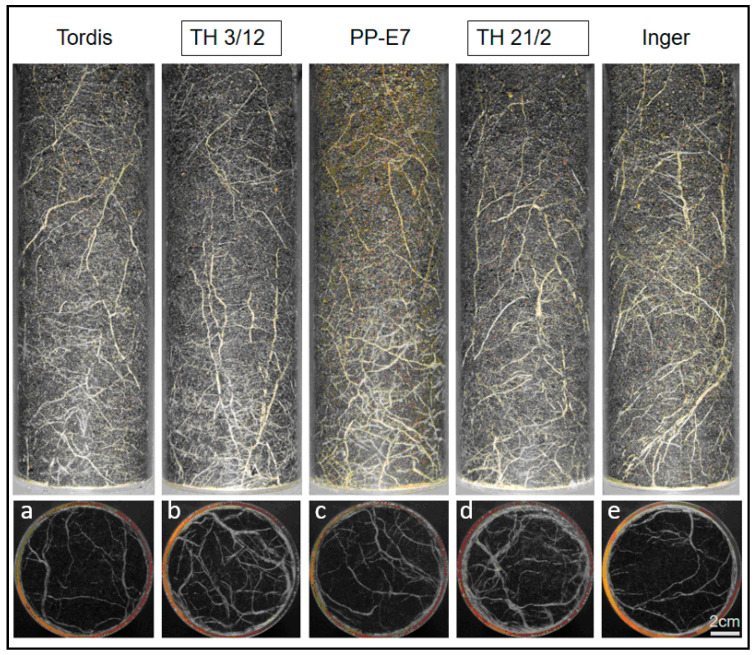
Root growth comparison of hybrid and parental plants. (**a**–**e**) Side and bottom views of roots represent root systems of willow plants grown in soil in transparent wall Plexiglas columns. (**b**,**d**) The triploid hybrids (TH3/12, TH21/2) produced an enlarged root system with thicker roots relative to both of the diploid cultivars (Tordis, Inger in panels (**a**,**e**)) and the autotetraploid male parents (**c**). Digital images were taken after eight weeks of cultivation. Bar is 2 cm for all images.

**Figure 3 genes-14-01929-f003:**
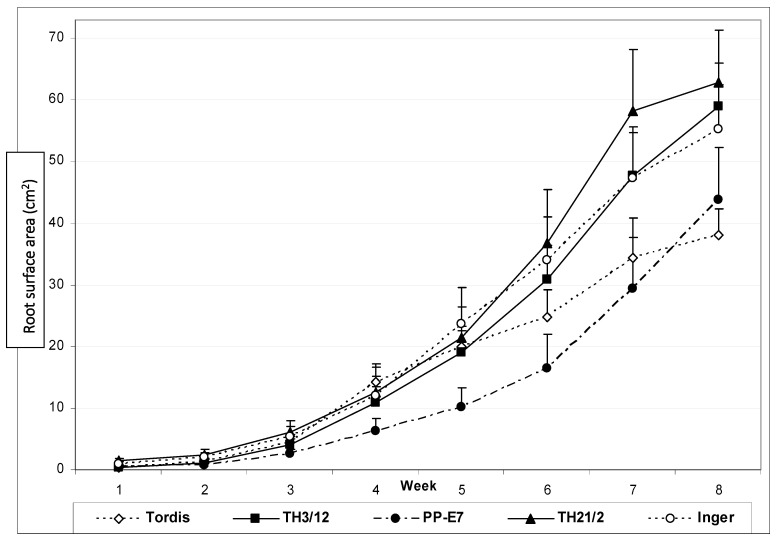
Differential expression of genotype-dependent root formation potential during the growth period of eight weeks in soil. Comparison of white pixel-based root surface area monitored by digital photography to record the root biomass growth of energy willow triploid hybrid (TH) plants and tetraploid parental plants (PP-E7) in comparison with roots of the diploid commercial cultivars (Inger and Tordis).

**Figure 4 genes-14-01929-f004:**
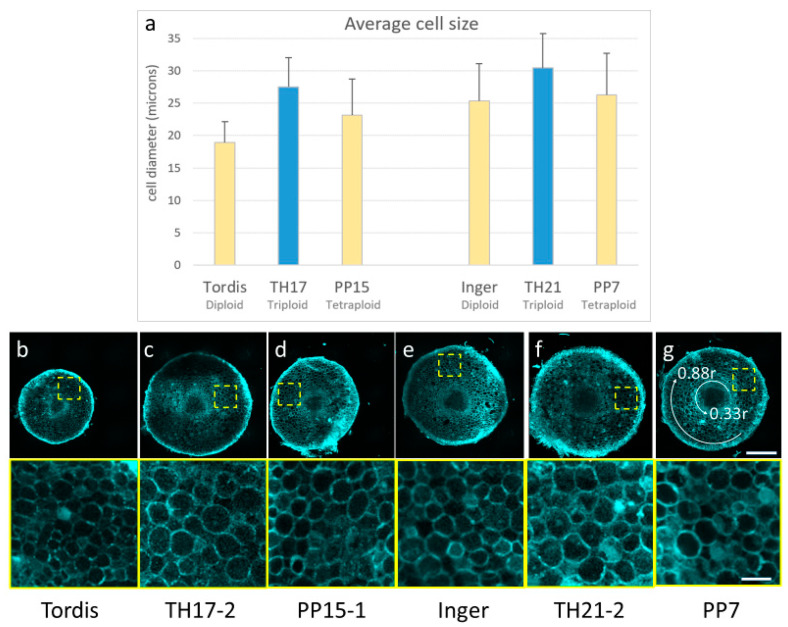
Differences in cell size of root parenchyma cells of triploid hybrid plants (TH) compared to diploid (Tordis, Inger) or tetraploid (PP-E15) plants. (**a**) Average cell diameter of root parenchyma cells measured on hand-sections prepared from root tips of indicated plants. (**b**–**g**) Representative confocal microscopy images of root disks displaying intrinsic cell wall fluorescence upon violet laser excitation (top panel). Bottom panel shows close-up images of the indicated rectangular ROIs. The last panel shows the position of 0.33r and 0.88r circles between which the parenchyma cell measurements were taken. Scale bars are 250 µm (top) and 50 µm (bottom).

**Table 1 genes-14-01929-t001:** Triploid hybrid willow plants show heterosis in soil-grown root biomass produced in relation to parental combinations.

Genotypes	Root Surface Area (cm^2^)	MPH%	Root Weight (g)	MPH%
Tordis	37.99 ± 4.35 ***		8.89 ± 0.92 ***	
**TH 3/12**	**58.92 ± 7.08**	**43.99**	**13.93 ± 1.92**	**19.31**
PP-E7	43.85 ± 8.43 ***		14.46 ± 1.88	
Tordis	37.99 ± 4.35 ***		8.89 ± 0.92 ***	
**TH 17/17**	**33.15 ± 16.26**	**−23.49**	**11.56 ± 1.57**	**9.63**
PP-E15	48.67 ± 7.92 **		12.20 ± 0.85	
Inger	55.29 ± 7.19 **		14.90 ± 2.02	
**TH 21/2**	**62.92 ± 8.44**	**26.93**	**16.15 ± 1.64**	**10.01**
PP-E7	43.85 ± 8.43 ***		14.46 ± 1.88 *	

Total surface area (in cm^2^) occupied by white pixels at the end of growing period (8th week) are presented to detect triploid heterosis in root biomass production. The TH3/12 hybrids show statistically significant heterosis relative to both parents (***, *p* < 0.01) and roots of the TH21/2 hybrids produced more roots than their parent PP-E7 (***, *p* < 0.01) or Inger (**, *p* < 0.05). Root weight parameters also indicate MPH% with statistically significant levels in TH3/12 and TH17/17 hybrid roots relative to Tordis roots (***, *p* < 0.01). In the case of the TH21/2 hybrid, the plants show significant heterosis in comparison to roots of the tetraploid PP-E7 plants (*, *p* < 0.1). The Mid-Parent Heterosis (MPH%) was calculated as heterosis over mid-parent (MPH%) = [(F1 − MP)/MP × 100], where F1 is the numerical value trait measurement in the hybrid and MP values are the mean values of the parents (P1 + P2)/2. Data for triploid hybrids are highlighted in bold for better clarity.

**Table 2 genes-14-01929-t002:** Monitoring the expression of triploid heterosis in root length, growth intensity, and frequency of G2/M cells in Knop’s solution.

GENOTYPES	ROOT GROWTH in Knop’s Solution			
	Primary Root Length after 16 Days (mm)	Root Growth Rate (mm Growth in the Last 48 h)	MPH (%) CPH (%)	Frequency of G2/M Cells (%) and MPH (%)
Tordis	79.20 ± 20.73 *	13.90 ± 9.06 ***		25.57
**TH3/12**	**64.71 ± 15.28**	**42.86 ± 7.99**	**70.08** **208.35**	**15.24** **−17.75**
PP-E7	110.33 ± 28.55 ***	36.50 ± 13.49		11.49
Tordis	79.20 ± 20.73	13.90 ± 9.06 **		25.57
**TH17/17**	**81.23 ± 31.27**	**21.77 ± 10.62**	**0.00** **56.62**	**15.70** **−14.97**
PP-E15	91.92 ± 20.90	36.92 ± 14.56 ***		11.36
Inger	55.36 ± 24.03 ***	19.18 ± 8.48 **		21.30
**TH21/2**	**106.42 ± 33.46**	**30.67 ± 12.49**	**10.17** **59.91**	**16.85** **2.78**
PP-E7	110.33 ± 28.55	36.50 ± 13.49		11.49

Tetraploid plants, especially from PP-E7 genotypes, developed longer roots between diploid Inger and TH21/2 hybrids (***, *p* < 0.01), between diploid Tordis and TH3/12 hybrid plants (*, *p* < 0.1), or between TH3/12 hybrids and PP-E7 plants (*** *p* < 0.01). In root growth rate values, statistically significant events were detected by comparison between diploid Inger and TH21/2 hybrids (**, *p* < 0.05), diploid Tordis and TH3/12 hybrid roots (***, *p* < 0.01), TH17/17 hybrids and diploid Tordis roots (**, *p* < 0.05), or TH17/17 hybrids and tetraploid PP-E15 plants (***, *p* < 0.01). The Mid-Parent Heterosis (MPH%) was calculated as heterosis over mid-parent (MPH%) = [(F1 − MP)/MP × 100], where F1 is the numerical value trait measurement in the hybrid and MP values are the mean values of the parents (P1 + P2)/2. We also estimated heterosis over the cultivar parent (CPH%) = [(F1 − CP)/CP × 100]. Data for triploid hybrids are highlighted in bold for better clarity.

**Table 3 genes-14-01929-t003:** Hormonal status of root tip regions/energy willow root tip regions with different genetic origins.

PARENTS	TORDIS		PP-E7		INGER	TORDIS		PP-E15
TRIPLOID HYBRIDS	→	TH3/12	← →	TH21/2	←	→	TH17/17	←
Indole-3-acetic acid (IAA)	463.29	**166.12**	211.37	**251.35** *****	280.89	463.29	**226.12**	321.52
IAA-glutamate MPH (%)	66.41	**125.20** **153.44**	32.39	**92.59** **158.92**	39.13	66.41	**24.69** *****	17.36
Phenylacetic acid	40.97	**34.69** *****	29.69	**29.67** *****	31.95	40.97	**42.20** *****	148.83
Salicylic acid	1031.56	**480.20**	973.82	**729.02**	769.18	1031.56	**694.92**	728.84
Benzoic acid	2097.12	**1439.9** *****	1088.23	**1628.9** *****	1717.8	1034.56	**1194.1** *****	3054.00
Jasmonic acid	331.92	**220.51** *****	211.98	**193.98**	705.62	331.92	**235.64**	318.2
Gibberellin_19_ MPH (%)	15.73	**78.07** **99.95**	62.36	**56.29** *****	54.17	15.73	**22.48** *****	33.58
Abscisic acid (ABA) MPH (%)	26.39	**27.89** **11.87**	23.47	**16.15**	18.40	26.39	**16.32**	21.43
Dihydro- phaseic acid MPH (%)	28.87	**31.76** **28.50**	20.56	**21.66** **2.34**	21.77	28.87	**16.82**	20.56
Phaseic acid MPH (%)	29.80	**87.30** **148.93**	40.34	**50.97** **26.63**	40.16	29.80	**24.81**	44.40
ABA-glucose ester	0.00	**1.82**	0.00	**5.62**	0.00	5.28	**0.00**	2.05
*Trans*-zeatin	11.57	**11.67**	24.98	**12.98**	9.12	11.57	**2.28**	9.85
Cytokinin (CK) ribosides MPH (%)	18.42	**26.59** *****	38.86	**16.44** *****	14.93	18.42	**19.40** **5.55**	18.34
CK O-glucosides	5.13	**2.06**	2.65	**2.28**	4.96	5.13	**1.77**	8.22
*Trans*-zeatin riboside monophosphate MPH (%)	6.46	**13.36** **18.81**	16.03	**17.81** *****	29.82	6.46	**11.32** **2.26**	15.68
Isopentenyl adenosine monophosphate MPH (%)	19.22	**50.18** **67.10**	40.84	**43.30** **6.57**	40.42	19.22	**35.86** **21.33**	39.89
*Cis*-zeatin riboside monophosphate MPH (%)	11.10	**15.44** **112.53**	3.43	**4.45** *****	8.42	11.10	**9.74 ** **20.25**	5.10
CK phosphates MPH (%)	36.78	**78.98** **62.71**	60.30	**65.56** *****	78.65	36.78	**56.92** **16.82**	60.67
Total CK MPH (%)	71.90	**119.30**	126.79	**162.89** **45.36**	97.33	71.90	**80.70**	97.08

Stem cuttings were rooted in Knop’s solution for 2 weeks, and 5 to 8 mm root tips were collected for hormone analyses. The values presented show the amounts of different hormones as pmol g^−1^ fresh weight. Mixtures of root tips from three plants were analyzed for each genotype. Asterisks (*) indicate intermediate hormone content for the TH roots between the parental values. Arrows indicate the parents for triploid lines. Data for triploid hybrids are highlighted in bold for better clarity.

## Data Availability

Not applicable.

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
