# Peer review of "Manifestation of Triploid Heterosis in the Root System after Crossing Diploid and Autotetraploid Energy Willow Plants"

_genes, 2023, doi:10.3390/genes14101929_

Round 1

Reviewer 1 Report

Main comments:

Rooting willow for energy purposes is the basis for intensifying the cultivation of this plant. Physiological aspects, including the concentration of hormones, play a special role in understanding the functioning of this plant and the intensity of growth of both roots and above-ground parts. The authors of the manuscript planned and performed very interesting and practically useful research. Genetic determinants of both abiotic and biotic stresses are currently a source of intense research by scientists.

The authors perfectly prepared the research material and the results obtained. Documenting appropriately in the form of detailed photographs and appropriate tables. The manuscript is well written. In the Introduction, the authors explained in detail the subject of the research, based on information collected from well-selected scientific journals. They provided detailed information in Materials and Methods. In the Results chapter, they skillfully described the obtained data and in the Discussion, they made good references to the research of other authors. For this reason, the manuscript with minor additions is worth publishing.

Detailed comments and suggestions:

Materials and Methods:

Line 127- 128: I suggest using the term: Knop's solution (also in other places in the text)

Line 135: Please provide the dimensions (capacity) of the plexiglass columns used for growth (diameter, length, material thickness). Lower case: plexiglass columns.

Line 139: The root intensity or density? In my opinion, one term (density) should be used throughout the manuscript to describe visible root surfaces.

Results

Line 208: Figure 1. Improve text sharpness in graphs.

Line 219: Figure 2 (not italics). Root growth comparison (root density)

Line 225: Total root surface area values ​​(root density) also Line 230.

Table 1: instead of intensity, use: density. Units (%, g) under the term as in the second column.

Table 3/A: second row IAA (Indole-3-acetic acid), last ABA (Abscisic acid)

Table 3/B: Third row, provide full name: CK.

Discussion

Line 439: It should be: Root growth intensity.

Author Response

Dear Reviewer,

Thank you very much for taking the time to review our manuscript. Please find the detailed responses below and the corresponding revisions/corrections highlighted in the re-submitted file.

Thank you for pointing out to the following details, we hope that the corrections made clearer our manuscript!

  1. ˝Line 127- 128: I suggest using the term: Knop's solution (also in other places in the text)˝

we have corrected it in each place in the text.

  1. Line 135: Please provide the dimensions (capacity) of the plexiglass columns used for growth (diameter, length, material thickness). Lower case: plexiglass columns.

We gave the dimensions of our plexiglass columns and corrected it for lower case.

  1. Line 139: The root intensity or density? In my opinion, one term (density) should be used throughout the manuscript to describe visible root surfaces.

We corrected intensity and density to surface area throughout the manuscript to describe visible root surfaces, as we determined it by summarizing the areas of white pixels.

  1. Line 208: Figure 1. Improve text sharpness in graphs.

We have improved the text sharpness on Figure 1. For publishing the manuscript we will prepare new Figure with high resolution graphs.

  1. Line 219: Figure 2 (not italics). Root growth comparison (root density)
  2. Line 225: Total root surface area values ​​(root density) also Line 230.

We would prefer to leave these as we wrote originally, based on our decision written in 3. point.

  1. Table 1: instead of intensity, use: density. Units (%, g) under the term as in the second column.

The surface area was determined; therefore, we feel that it is better to use it in our article instead of intensity and density.

Table 3/A: second row IAA (Indole-3-acetic acid), last ABA (Abscisic acid)

We have given the abbreviation of Indole-3-acetic acid in the second row, and Abscisic acid in the last one. We hope that it helps in understanding.

Table 3/B: Third row, provide full name: CK.

We gave the full name of CK in the third row. We hope that it helps in understanding.

Discussion

Line 439: It should be: Root growth intensity.

We corrected the subtitle to the following:

4.2. Root growth intensity is a primary factor in generation of triploid heterosis in root systems

If necessary, we are open to further modifications.

Best regards,

Györgyi Ferenc

Reviewer 2 Report

Comments and Suggestions for Authors

The present study is focused on the characterization of triploid heterosis controlling biomass, growth rate, cellular parameters, and hormonal status of roots of energy willows. For this purpose, triploid hybrids were produced by appling polyploidy and heterosis effects and consecutively analysis of consequences in root development was carried out.  Shrub willows grown as woody crop have outstanding potential to serve as an optimal feedstock produce bioenergy, biofuels, and bioproducts with environmental and rural development benefits. The environmental and the economical sustainability of the above listed multipurpose applications exploits some key properties of energy willow genotypes that can be generated by improved breeding technologies

The structure of manuscript meets the requirements of “Genes”. The research methods applied are appropriate and sufficient to achieve the objectives of the study. The results are clearly presented and supported by tables and figures of good quality, and statistical analysis.

The following recommendations can be made:

Introduction:

The part between lines 38-66 relating to the ecological and economic use of energy willow (Salix ssp.) should be shortened by briefly providing information on these benefits as the focus of the present study is otherwise.

Materials and Methods:

Тhе point 2.2. Flow cytometry have to begin with the sentence “Determination of ploidy levels of obtained hibrids was conducted by flow cytometry …..”

 Results

In the tables, the explanations should be removed from the titles and placed below the tables as footnotes.

Discussion

Subheadings in the “Discussion” sound like results - to be edited

 Conclusions

The two first sentences of this section are too common and can be removed.

In conclusion, this manuscript is recommended for publication in “Genes”.

Author Response

Dear Reviewer,

Thank you very much for taking the time to review our manuscript. Please find the detailed responses below and the corresponding revisions/corrections highlighted in the re-submitted file.

Thank you for pointing out to the following details, we hope that the corrections made clearer our manuscript!

Introduction:

The part between lines 38-66 relating to the ecological and economic use of energy willow (Salix ssp.) should be shortened by briefly providing information on these benefits as the focus of the present study is otherwise.

We have shortened the first 3 paragraphs of the introduction (see the highlighted lines)

Materials and Methods:

Тhе point 2.2. Flow cytometry have to begin with the sentence “Determination of ploidy levels of obtained hibrids was conducted by flow cytometry …..”

We have inserted the suggested sentence, see it highlighted in the text.

 Results

In the tables, the explanations should be removed from the titles and placed below the tables as footnotes.

Thanks for your suggestion, we have modified the tables.

Discussion

Subheadings in the “Discussion” sound like results - to be edited

We have edited the subheadings in the Discussion.

 Conclusions

The two first sentences of this section are too common and can be removed.

We have removed the first two sentences.

If necessary, we are open to further modifications.

Best regards,

Györgyi Ferenc
